# Medical Cost of Cancer Care for Privately Insured Children in Chile

**DOI:** 10.3390/ijerph18136746

**Published:** 2021-06-23

**Authors:** Florencia Borrescio-Higa, Nieves Valdés

**Affiliations:** Business School, Universidad Adolfo Ibáñez, Santiago 7941169, Chile; florencia.borrescio@uai.cl

**Keywords:** childhood cancer, medical cost, administrative data

## Abstract

Medical care for children with cancer is complex and expensive, and represents a large financial burden for families around the world. We estimated the medical cost of cancer care for children under the age of 18, using administrative records of the universe of children with private insurance in Chile in the period 2007–2018, based on a sample of 3853 observations. We analyzed total cost and out-of-pocket spending by patients’ characteristics, type of cancer, and by service. Children with cancer had high annual medical costs, USD 32,287 on average for 2018. Costs were higher for the younger children in the sample. The vast majority of the cost was driven by inpatient hospital care for all types of cancer. The average total cost increased 20% in real terms over the period of study, while out-of-pocket expenses increased almost 29%. Private insurance beneficiaries faced a significant economic burden associated with medical treatment of a child with cancer. Interventions that reduce hospitalizations, as well as systemwide reforms that incorporate maximum out-of-pocket payments and prevent catastrophic expenditures, can contribute to alleviating the financial burden of childhood cancer.

## 1. Introduction

Cancer is the second-leading cause of death for children ages 5 to 15 in Chile [1]. For this age group, the age-standardized incidence rate for leukemia (the most common type of cancer) is higher in Chile than the average in Latin America, and is also higher than the average for other high-income countries (4.9 vs. 4.1 and 4.1 per 100,000 persons/year, respectively), whereas the associated mortality rates are lower than in Latin America, but still higher than in high-income countries (1.4 vs. 2.1 and 0.61 per 100,000 persons/year, respectively) [2].

Childhood cancer survival rates have increased dramatically in recent decades, mainly due to advances in diagnosis and treatment [3,4]. The treatment of cancer in pediatric patients is complex, costly, and resource-intensive, and is even more expensive than the treatment for adults [3,5,6]. The financial expenses encompass the associated cost of treatment, diagnostic assessments, and hospitalization, as well as the cost of prescription drugs. Moreover, the costs of cancer care persist beyond the treatment and can be long-lasting, because individuals with a cancer history face higher costs for medical care [7].

Further, the costs of cancer care have increased in the past 10 years, and are likely to continue this trend [8,9]. The high cost, through copayments and out-of-pocket (OOP) expenditures, can pose a significant barrier to receiving timely and appropriate medical care, which can have an impact on treatment outcomes and can lead to treatment abandonment [10,11]. In this context, both the access and the affordability of treatment are key concerns [12]. Although the uninsured face obstacles in accessing care and are often diagnosed at later stages [4], individuals who are fully insured or with private insurance often face high out-of-pocket costs, which can create major financial hardship [13,14,15].

Most available evidence on the costs associated with childhood cancer is concentrated in developed countries [6,15]. For example, Mueller et al. analyzed utilization and spending among children with cancer enrolled in Medicaid [16], and Warner et al. analyzed the financial burden of cancer patients under the age of 21 based on a single-site, cross-sectional survey in the US [17]. Other recent examples include de Oliveira et al., who studied public-payer healthcare costs by phase of cancer care in one province in Canada [5], and Chae et al., who focused on the financial cost within a universal health insurance system (excluding private insurance) in Korea [18].

Most available evidence for Chile is focused on adults [19,20,21], with some exceptions. Using data for six hospitals in Santiago, Chile, Santolaya et al. showed that for children with febrile neutropenia, early discharge with antibiotics at home cost less than continuing in the hospital [22]. However, no systematic study exists for this country regarding the medical costs that families with a pediatric cancer patient face.

Chile has a mixed, non-complementary public–private (both in provision and insurance) healthcare system, in which workers choose the National Health Fund or a private insurer for their mandatory contributions. Health insurance coverage is large, given that 95.2% of the population is insured [23]. However, coverage is small in terms of the percentage of services and cost coverage: OOP expenditure as a share of total health expenditure is 33%, which is one of the highest among OECD countries [24].

In response to disparities in access, care, and quality of medical treatment, the national health system underwent a major reform in 2005, to provide guaranteed healthcare for those suffering from catastrophic illnesses, including all types of childhood cancer [25]. This reform had a positive impact in terms of coverage and equity in healthcare [21].

In terms of cancer treatment, the reformed system implies private health insurance beneficiaries have two options for the medical care of pediatric cancer. They can rely on their insurance and pay for each healthcare event in accordance with their contract, or they can use a DRG (diagnosis-related group)-based subsidized payment system that has a pre-determined healthcare provider (usually different from the one attached to their insurance contract) and a fixed coinsurance, called GES (acronym for the Spanish “Garantías Explícitas en Salud”) [26,27].

The objective of this paper is to examine the annual medical cost of cancer care for children under the age of 18. We base our analysis on administrative records of the universe of patients enrolled in private health insurance in Chile between 2007 and 2018. Note the analysis is based on individuals with private health insurance, because comparable data for individuals with public insurance do not exist. This paper contributes to the literature by analyzing in a systematic way the medical costs faced by privately insured pediatric cancer patients’ families in a multi-institution study over a 12-year span, which has not been done before in this setting. Although results cannot be extrapolated to beneficiaries of public health insurance, some of our main findings can be relevant in a broader setting, because clinical scenarios for children with cancer are generally independent of the type of insurance [16].

Our results show the financial costs of cancer care for pediatric patients can be substantial. Children with cancer have high annual medical costs, and costs are higher for the younger children in the sample. We also find the vast majority of the cost is driven by inpatient hospital care for all types of cancer. Moreover, the average total cost increased 20% in real terms over the period of study, while OOP expenses increased almost 29%.

## 2. Materials and Methods

### 2.1. Study Population

The population of interest for our study included 0- to 17-year-olds who had at least one non-newborn hospitalization with a discharge diagnosis of cancer in a given year, and who were enrolled in private health insurance. Although the private insurance system covers roughly 16% of the Chilean population, data for 2018 show privately insured children represented 42% of hospitalized pediatric patients who faced some kind of copayment or OOP expenses, because this category includes patients enrolled in the top tiers of public insurance [28,29].

### 2.2. Data

We used the administrative records of the Superintendencia de Salud de Chile, the institution of the central government in charge of monitoring the private health insurance system. For each year from 2007 to 2018, information was available on insurance contracts, including socio-demographic information on beneficiaries, and claims data, which contained the costs of all individual claims associated with each event of utilization of the insurance. We combined information to build a data set in which each observation was a child under the age of 18. To do so, we first identified all children who had at least one non-newborn hospital stay. We classified as newborn all hospital stays with a discharge date in the same month as the child’s birth.

In addition, and crucially for our purposes, we identified children as having cancer if they received at least one discharge diagnosis code of cancer following a hospitalization in a given year, because administrative records do not include medical history. We classified cancers using the International Classification of Childhood Cancer, which groups ICD-10 codes as follows: leukemia (C91–C95), brain and central nervous system cancer (C70–C72), bone and articular cartilage cancer (C40–C41), non-Hodgkin’s lymphoma (C82–C85, C96), mesothelioma soft tissues (C45–C49), and groups all of the remaining types of cancers in an “other cancer” category [30]. For children with more than one hospital stay in a given year, we used the most frequent type of cancer diagnosis.

We obtained demographic information for all beneficiaries from the insurance contracts, including date of birth and gender, and additional information for the policyholder, such as monthly income and municipality of residence. Some children in the working sample had different policyholders (and health insurance contracts) in different months of a given year. For these children, we used the most frequent policyholder’s date of birth, gender, municipality of residence, and income.

We matched healthcare utilization, including the type of care (inpatient and outpatient) and the related costs (total, OOP, and by type of care), to each child. We computed total annual costs by adding all payments the insurance company and the policyholder made in a given year to the healthcare providers where the child received treatment. We computed annual OOP expenses as the sum of payments made by the policyholder. We converted all monetary variables (income and costs) to constant 2018 dollars using the price index and the exchange rate published by the Central Bank of Chile [31]. Finally, we included in our analysis an indicator variable that took a value of 1 if the child had at least one claim reimbursed under the publicly subsidized DRG payment system in a given year.

### 2.3. Sample Selection

Pooling administrative records from 2007 to 2018, we had 4303 observations (child-year) of children with at least one non-newborn hospital stay with a discharge diagnosis of cancer in a given year. We made several restrictions to the data set: (1) we kept children who were beneficiaries of private health insurance during all 12 months of the year, or since birth for those less than one year old by December (99.86% of observations); (2) we kept children for whom we had complete information on costs (90.43% of observations); and (3) we excluded extreme values by eliminating observations with costs in the upper 1% (0.99% of observations with complete information on costs). All results are qualitatively similar with different thresholds for the trimming. Results available upon request. The final sample included 3853 observations (child-year) with an average of approximately 321 children with cancer per year.

### 2.4. Statistical Analysis

We used a sample of children with at least one non-newborn hospital stay with a discharge diagnosis of cancer in a year to study them in two dimensions: their demographic and economic characteristics, and the utilization of healthcare services and related costs.

We first conducted a comparison of demographic and economic characteristics at the beginning and at the end of the study period, performed t-tests of differences in means, and reported the corresponding *p* value of the significance of the difference using a bilateral alternative hypothesis, assuming unpaired data had equal variances in both groups.

We then analyzed the total and OOP medical costs by type of healthcare services. We found trend patterns between 2007 and 2018 and differences across types of cancer. Finally, we found evidence on the association between medical costs (total costs and OOP payments, and by type of care, inpatient and outpatient) and a series of covariates including child and policyholder’s information, and year fixed effects to control for factors changing each year that were common to all patients in a given year.

We estimated the parameters of generalized linear models (GLMs) by quasi-maximum likelihood (QML), clustering standard errors at the policyholder’s identification number level [32]. GLMs are appropriate to estimate healthcare costs because they are able to predict outcomes that are highly skewed, to directly model heteroscedasticity, and perform well for outcomes with mass at zero (such as OOP payments) [33,34,35,36,37]. GLMs are more flexible than ordinary linear models because they allow the expectation of the outcome to be a non-linear function (known as the link function) of the linear index of covariates, and allow the variance to be a function of the expectation through a distribution family. As a result, the implementation of a GLM requires the choice of a link function and a distribution family. We validated the use of the log link function and the gamma distribution family using a combination of information criteria, Box–Cox approach, and modified Park tests, as proposed by Deb et al [38]. Details of these choices are presented in Appendix B.

A GLM with log link and gamma distribution is characterized by the following conditional moments:$$E\left(y_{it}|X_{it}\right)=\exp\left(X\prime_{it}\beta \right)\;\mathrm{and}\;v(y_{it}|x_{it})\approx {\left[E\left(y_{it}|X_{it}\right)\right]}^{2},$$
where *i* indexes children and *t* indexes years.

We estimated the model for six different outcomes, *y_it_*: total healthcare costs, inpatient care costs, outpatient care costs, OOP payments for all care, OOP payments for inpatient care, and OOP payments for outpatient care. Covariates in *X_it_* included the following: (a) child’s age (a set of dichotomous variables with the age interval (0–4) as the reference category); (b) child’s gender (a binary variable equal to 1 if the child was male); (c) type of cancer (a set of dichotomous variables with “other cancer” as the reference category); (d) type of payment scheme (a binary variable equal to 1 if the child used the GES payment scheme); (e) age and age squared for the policyholder; (f) policyholder’s gender (a binary variable equal to 1 if the policyholder was male); (g) region of residence (a binary variable equal to 1 if the policyholder resided in the Metropolitan area); (h) policyholder’s income quantiles (a set of dichotomous variables with the 5th quantile as the reference category); and (i) year fixed effects (a set of dichotomous variables reflecting each year of data with 2007 as the reference category).

### 2.5. Patient Involvement

We were granted access to de-identified administrative records of the private health insurance system by the Superintendencia de Salud de Chile. As a result, patients were not involved in this study.

## 3. Results

### 3.1. General Characteristics of the Sample

Table 1 presents descriptive statistics of the sample and shows a cancer diagnosis was more common among children between the ages of five and nine. Gender appeared to be fairly balanced, and we found that leukemia was the most common type of cancer in both years.

We found no statistically significant differences between 2007 and 2018 in almost all variables analyzed. The exceptions were the percentage of children with bone or articular cancer, the percentage of children who used the GES payment system at least once in the year, and the policyholder’s average annual income (*p* = 0.004, *p* < 0.001, and *p* < 0.001, respectively).

### 3.2. Temporal Trends in Pediatric Healthcare Costs

Between 2007 and 2018, average total costs increased 20.3% in real terms (i.e., in constant 2018 dollars), as shown in Figure 1. Average OOP payments represented between 19% 26% of total costs in the period analyzed.

Moreover, the financial burden of the treatment of cancer intensified its impact on the patients’ families, as average OOP payments increased 29.3% in the same period.

### 3.3. Temporal Trends in Total Costs and Out-of-Pocket Payments by Type of Healthcare Service

Most of the expenses, total and OOP, corresponded to the utilization of inpatient healthcare services (see Figure 2). For example, inpatient costs represented 92.6% of total and 87.2% of OOP payments in 2018. Inpatient total costs increased 20% over the period, while inpatient OOP expenses increased 27.45%.

### 3.4. Cost of Care by Type of Cancer

Children with a diagnosis of leukemia faced the most expensive costs for medical care. The average total costs for this type of cancer were the highest (USD 46,309), and were 2.5 times the average costs of cancer care in the “other cancer” category (USD 19,051) (see Figure 3). However, OOP payments that families of children with leukemia faced were the third highest, after cancer of the brain or nervous system and of bones or articular cartilage. In terms of OOP payments, mesothelioma and soft-tissue cancer were the least costly types of cancer.

The share of average inpatient and outpatient care costs over average total costs and OOP payments were highly homogeneous across different types of cancer (on average, inpatient care represented 92.7% of all costs and 87.2% of OOP spending). The exception was the distribution of out-of-pocket costs for children with mesothelioma or soft-tissue cancer. For this category, inpatient and outpatient care costs represented 83% and 17%, respectively, of average OOP payments.

### 3.5. Regression Analysis of Childhood Cancer Medical Costs

Table 2 shows marginal effects of several patient and policyholder characteristics on medical costs, both total and OOP, and by type of care (all care, inpatient, and outpatient). Considering the statistically significant estimates (at the 1% level) reported in columns (1) and (4), we found evidence that medical care was more expensive for the youngest patients (under five years old) and for patients with leukemia. In particular, total costs among children between ages five and nine were USD 13,970 lower than for those under five (*p* < 0.01). Since the average total cost was equal to USD 32,124, this implied a difference of approximately 43%. Families of children with leukemia faced costs that were USD 28,305 higher than those for cancers included in “other cancer” diagnoses (*p* < 0.01), a difference of approximately 88% from the average total cost. Families of children with leukemia or cancer of the brain or nervous system faced OOP costs that were USD 4369 and USD 5972 higher, respectively, than for cancer care in the “other cancer” category (and 60% and 83% higher than the average OOP of USD 7236).

Further, total costs were positively associated with residence in the Metropolitan region (where the country capital is), where policyholders paid USD 3211 more (approximately 10%) for their dependents’ cancer care than those living in other regions of Chile (*p* < 0.05). Total costs also were higher among patients who used the DRG payment system, GES. In addition, policyholders with income in the second quantile faced OOP expenses that are USD 3008 lower than those in the fifth quantile (*p* < 0.01), and approximately 42% lower than the average OOP.

Differences emerged in marginal effects when we compared inpatient with outpatient total costs (columns (2) and (5)) and inpatient with outpatient OOP payments (columns (3) and (6)). Marginal effects for inpatient care were very similar to those obtained considering all types of care. The opposite was true for outpatient care: outpatient care costs were higher for adolescents between 15 and 17 years of age, and patients with cancer of the brain or nervous system made the highest total and OOP payments for outpatient care.

Among patients who used the DRG payment system (GES), total costs, inpatient costs, and outpatient costs were USD 8882 (*p* < 0.01), USD 8352 (*p* < 0.01), and USD 376 (*p* < 0.05) higher, respectively, than the cost of those who relied on their insurance (and 27.7%, 28%, and 15.7% higher than the corresponding average total cost). However, OOP payments related to all care, inpatient care, and outpatient care among GES users were USD 3363, USD 3128, and USD 241 lower than for comparable patients who did not use GES (*p* < 0.01). Considering the average OOP payments for each type of care, these implied a reduction of approximately 46.5%, 49%, and 28.5%, respectively.

We present coefficient estimates used to compute marginal effects in the Appendix A.

## 4. Discussion

Whereas in some areas of Latin America, a lack of adequate hospital infrastructure and expertise means pediatric cancers are not effectively managed, in recent years, Chile has implemented comprehensive, meaningful changes to improve cancer care for children, in particular with the implementation of a population-based registry [29,39]. The “Plan Nacional de Cancer” (National Cancer Plan was launched in 2018 and marked an unprecedented relevance of cancer in the national health policy, not only considering it a health condition, but also taking into account its social and economic dimensions in the design of a plan of several actions for the period 2018–2028 [40,41].

Generally, childhood cancers are highly curable, but effective management is complex and expensive [3,5,6,29]. For the period 2007–2011, the five-year survival rate among children under the age of 15 was 71.4% [1]. However, information on the medical costs faced by families with a child who has cancer is lacking. Accurate measures of costs in different settings are essential, because when healthcare providers receive information on costs, they will likely be more mindful of their decisions. Additionally, our findings will inform future steps in the national policy planning for improving cancer care for children [42].

Using administrative records of patients with private health insurance in Chile between 2007 and 2018, we examined the annual medical cost of cancer care for children and adolescents. An advantage of these data is that they comprised all private insurance patients who accessed health care in all private hospitals and medical institutions in Chile.

We found the total medical cost of cancer care in 2018 was USD 32,287, which was lower than the cost reported in Mexico and similar to the one reported for Korea [4,18]. To put this figure in perspective, the total medical cost in 2018 was twice the size of per-capita GDP (gross domestic product), though this ratio was much lower than the one reported for Mexico, which was 6.3 times per-capita GDP in 2016 [4,43]. Average total cost increased 20% in real terms over the period of study, while OOP spending increased almost 29%, which put pressure on the question of affordability of cancer care and increased the danger of treatment abandonment [11,44,45]. On average, OOP expenses represented 24.5% of total costs, in a country where OOP expenditures as a share of total health expenditure (for all medical care) were relatively high, at 33% [24]. In 2018, the direct economic burden of pediatric cancer, represented by annual OOP payments for medical care, was approximately 28.9% of the annual income of the average policyholder. This was similar in magnitude to findings in a different setting [7,46].

The regression analysis revealed age was negatively associated with cost, because the youngest patients had the highest expenses. Leukemia was the most expensive cancer in terms of total cost, but cancer of the brain or nervous system had the higher OOP spending. Consistent with findings from other settings, leukemia was the most frequent type of cancer and had the highest average total cost [18,39,47,48]. However, we found cancer of the brain or nervous system, and bone and articular cartilage cancer, had the highest average OOP costs, whereas evidence from the US showed non-Hodgkin’s lymphoma was the most expensive category among children [16].

The vast majority of the cost was driven by inpatient hospital care for all types of cancer, for example, 77% of total costs and 73% of OOP payments in the case of leukemia. Children with cancer usually require high levels of inpatient care, which can include not only chemotherapy administration, but also unexpected hospitalizations to manage complications, which can involve relatively long hospital stays [16,17]. In this context, interventions that reduce hospitalizations in this population become crucial [22]. Additionally, implementing systemwide reforms that incorporate an OOP maximum may be also necessary in order to decrease the financial burden. The percentage spent OOP on outpatient care, between 11% and 19% depending on the type of cancer, was consistent with that reported in other settings [4,16].

Among patients who used the DRG-based payment system, GES, total costs were higher than for comparable patients who paid based on their insurance contract, but their OOP payments were lower. The GES program covered a gap for children with any type of cancer (with no limitations) who were underinsured with their current private insurance policy, and therefore, a minimum standard of care existed, particularly for lower-income individuals. This finding signaled that individuals were opting into the GES system when treatment was particularly expensive, as their OOP payments became lower. In this sense, one of the main objectives of the GES policy seems to be working in this setting, because patients can have access to expensive cancer treatment while being protected financially. Our results provided evidence that the GES policy has solved some issues and disparities in pediatric cancer care [25].

Recent calls to action for improvement in the care of children with cancer in low- and middle-income countries are related to improving access to medical care in two dimensions: implementation of effective childhood cancer services and enhancement of financial coverage [4]. All pediatric patients with cancer must have adequate insurance coverage, ensuring a minimum standard of care. The DRG-based payment system in Chile could be adapted to other settings to guarantee coverage, which would particularly benefit low-income patients.

Our study had certain limitations. The analysis was based on individuals with private health insurance, and therefore, results were inherent to the private health insurance system and cannot be extrapolated to beneficiaries of public health insurance. However, because clinical scenarios for children with cancer were generally independent of the type of insurance, some of our main findings, such as the high proportion of the cost driven by inpatient service utilization, could be relevant in a broader setting [16].

We were not able to capture the spending trajectory for pediatric cancer patients from the date of diagnosis through treatment, because we could not identify the date of initial diagnosis. The administrative nature of the data did not allow us to identify quality of care, length of stay, or fully quantify the impact of abandonment. Nonetheless, although analyzing health outcomes in relation to cost was not possible with the current data, survival rates reported for Chile were similar to those for developed countries [49].

Finally, the medical cost was not the only cost faced by families with a child who had cancer. Several other associated costs contributed to the overall financial burden of cancer [4,17,50,51]. This subject will be studied in future research.

## 5. Conclusions

The need to address the burden of childhood cancer is urgent, and despite vast heterogeneity across countries, one of the main barriers to cancer care is related to the medical cost of care. Both access and the affordability of treatment are key concerns. Although governmental actions in many Latin American and other low- and middle-income countries have launched nationwide cancer registries and have improved cancer care of children, many difficulties persist in terms of timely diagnosis, treatment provision, and prevention of abandonment.

We found private insurance beneficiaries incurred significant economic burdens associated with medical treatment of childhood cancer. Although access and coverage has improved through the publicly financed DRG payment system, the total medical cost of cancer care in Chile remains high, and inpatient hospitalizations are a key component of the high OOP expenditures. Interventions that reduce hospitalizations, as well as systemwide reforms that incorporate maximum OOP payments and prevent catastrophic expenditures, can contribute to alleviating the financial burden of childhood cancer.

## Figures and Tables

**Figure 1 ijerph-18-06746-f001:**
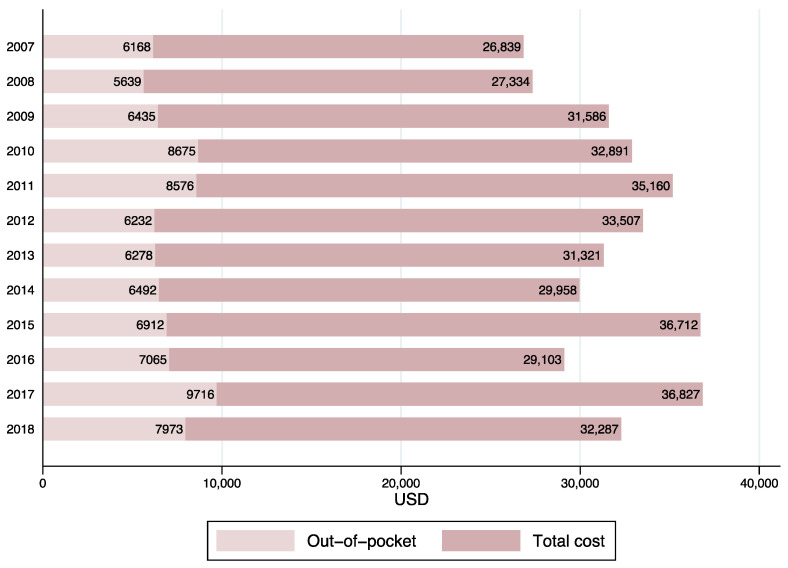
Average annual total costs and out-of-pocket payments, by year. Children with cancer, 2007 to 2018. Private health insurance system.

**Figure 2 ijerph-18-06746-f002:**
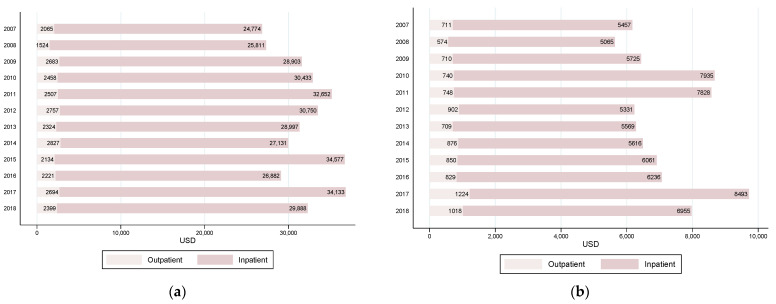
Inpatient and outpatient total costs and out-of-pocket payments. Children with cancer, 2007 to 2018. Private health insurance system. (**a**) Average total costs; (**b**) average out-of-pocket payments. For each year, inpatient and outpatient costs add up to total cost.

**Figure 3 ijerph-18-06746-f003:**
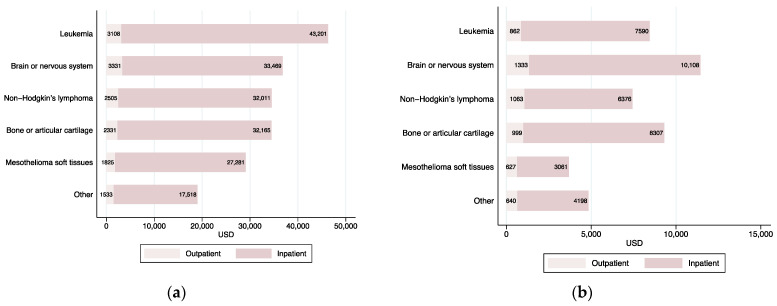
Inpatient and outpatient average annual and out-of-pocket costs, by type of cancer, 2007 to 2018. Private health insurance system. (**a**) Average total costs; (**b**) average out-of-pocket payments. For each year, inpatient and outpatient costs add up to total cost.

**Table 1 ijerph-18-06746-t001:** Descriptive statistics. Sample of children with at least one hospital stay with a discharge diagnosis of cancer, 2007 to 2018. Private health insurance system.

	All Years	2007	2018	Mean Difference *t*-Test *p*-Value: 2007 vs. 2018
*Patient*				
*n*	3853	248	407	
Distribution by age group (%)				
0–4	26.0	23.8	22.9	0.783
5–9	30.5	29.4	34.4	0.185
10–14	24.1	25.4	22.4	0.379
15–17	19.5	21.4	20.4	0.766
Mean age	8.8	9.2	8.9	0.478
% Male	52.8	51.2	52.8	0.689
Distribution by most frequent discharge’s diagnosis				
Leukemia	31.8	30.2	32.2	0.602
Brain or nervous system	14.0	15.3	14.3	0.709
Non-Hodgkin’s lymphoma	5.3	4.8	4.2	0.695
Bone or articular cartilage	4.9	10.5	4.2	0.004
Mesothelioma soft tissues	3.2	4.0	3.7	0.824
Other cancer	40.6	35.1	41.5	0.099
GES	48.8	28.2	49.4	<0.001
*Policyholder*				
Mean age	41.4	41.4	42.3	0.101
% Male	67.9	68.1	63.4	0.212
Distribution by policyholder’s region of residence (%)				
Metropolitan	58.9	57.3	62.7	0.173
South	20.7	25.0	17.7	0.029
North	20.4	17.7	19.7	0.541
Mean annual income	25,024	20,813	27,608	<0.001

Note: income and premium measured in constant dollars of 2018. Abbreviation: GES—Garantías Explícitas en Salud.

**Table 2 ijerph-18-06746-t002:** Marginal effects on annual total medical costs and out-of-pocket payments, by type of care. Children with cancer, 2007 to 2018. Private health insurance system.

	Total Costs			Out-of-Pocket		
	(1)	(2)	(3)	(4)	(5)	(6)
Variables	All	Inpatient	Outpatient	All	Inpatient	Outpatient
*Patient*						
Age 5–9	−13,969.85 ***	−13,991.49 ***	96.49	−3214.58 ***	−3220.38 ***	−57.85
	(2463.942)	(2435.629)	(188.172)	(671.170)	(665.850)	(77.940)
Age 10–14	−9674.74 ***	−10,245.84 ***	522.4 **	−1542.35 *	−1657.94 **	32.56
	(2767.828)	(2711.179)	(243.069)	(822.560)	(790.610)	(80.320)
Age 15–17	−9661.96 ***	−10,556.51 ***	661.95 **	−1496.72 *	−1669.15 **	51.2
	(3041.192)	(3023.489)	(262.877)	(856.600)	(829.360)	(84.810)
Male	1363.68	1304.63	94.78	−183.05	−233.59	11.92
	(1773.860)	(1747.659)	(169.036)	(546.240)	(523.200)	(58.310)
Leukemia	28,304.76 ***	26,939.25 ***	1353.22 ***	4369.01 ***	4153.19 ***	267.53 ***
	(2105.794)	(2100.442)	(215.345)	(705.640)	(692.950)	(62.310)
Brain or nervous system	18,485.8 ***	16,676.62 ***	1597.48 ***	5972.12 ***	5171.59 ***	688.78 ***
	(2644.383)	(2561.675)	(340.909)	(1179.220)	(1044.750)	(195.620)
Non-Hodgkin’s lymphoma	15,833.76 ***	15,170.36 ***	777.33 ***	3061.63 ***	2697.37 ***	427.54 ***
	(3906.076)	(3844.151)	(287.064)	(1087.530)	(1033.340)	(154.210)
Bone or articular cartilage	17,459 ***	16,742.74 ***	663.52 **	4070.47 ***	3909.27 ***	268.87 **
	(4115.738)	(4007.618)	(303.541)	(1491.240)	(1456.600)	(107.120)
Mesothelioma soft tissues	9162.15 **	8511.43 **	432.82	−750.11	−787.94	32.02
	(3655.754)	(3487.766)	(400.487)	(584.050)	(548.700)	(82.530)
GES payment	8882.34 ***	8351.97 ***	375.9 **	−3362.63 ***	−3127.56 ***	−241.39 ***
	(1782.42)	(1755.309)	(163.707)	(613.62)	(593.160)	(58.040)
*Policyholder*						
Residence in Metropolitan region	3210.62 *	3926.84 **	−539.1 ***	894.6	870.76	67.02
	(1838.65)	(1814.621)	(194.623)	(583.18)	(554.330)	(60.460)
Q1 of annual income	760.89	1839.36	−855.7 ***	−951.86	−344.02	−502.96 ***
	(3357.45)	(3320.530)	(277.783)	(1120.34)	(1078.720)	(126.120)
Q2 of annual income	−4803.83	−3856.67	−891.09 ***	−3007.65 ***	−2501.14 ***	−445.59 ***
	(3029.80)	(2969.836)	(251.228)	(888.16)	(821.320)	(125.280)
Q3 of annual income	−4744.52	−4241.48	−405.82	−1227.25	−917.27	−234.05 *
	(3111.73)	(3051.960)	(264.918)	(1036.66)	(974.580)	(130.890)
Q4 of annual income	−1816.63	−1710.82	−163.5	−1085.19	−839.01	−204.19
	(3217.22)	(3139.039)	(287.951)	(989.05)	(930.460)	(133.590)
Policyholder’s demographic controls	Yes	Yes	Yes	Yes	Yes	Yes
Year dummies	Yes	Yes	Yes	Yes	Yes	Yes
Observations	3644	3644	3621	3638	3212	3615
Mean of outcome	32,124	29,737	2387	7236	6388	848

Note: Marginal effects computed using QML estimates of GLM with log link and gamma distribution. Robust standard errors in parentheses, clustered at the policyholder’s id level. *** *p* < 0.01, ** *p* < 0.05, * *p* < 0.1. The reference category for type of cancer was “other cancer” diagnosis. We report QML estimates of GLM used to compute marginal effects in Appendix A. Abbreviations: GES—Garantías Explícitas en Salud; Q1, Q2, Q3, Q4—first, second, third, and fourth quintile of income distribution, respectively.

## Data Availability

The data that supported the findings of this study are available from the Superintendencia de Salud del Gobierno de Chile. Restrictions apply to the availability of these data, which were used under license for this study. Data are available from the authors only with the permission of Superintendencia de Salud del Gobierno de Chile.

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
