# Peer review of "Medical Cost of Cancer Care for Privately Insured Children in Chile"

_ijerph, 2021, doi:10.3390/ijerph18136746_

Round 1

Reviewer 1 Report

Cancer medical costs is a topic that has received a lot of attention.   The paper is generally well written and structured.  Minor revision is required as follows:

Introduction section:

  1. Please describe the government subsidy scheme, co-payment scheme and/or private insurance utilization and the change of the scheme, if any, in Chile. The change may affect the rising medical cost in 2017 as shown in the result section?
  2. Please describe the common types of pediatric cancer and its incidence and death rates in worldwide and in Chile.
  3. Please describe types of costs for cancer care and medical costs range in Chile and worldwide.

Table1: typo? 2018 instead of 2008? Or use incorrect data?

Discussion section:

  1. It seems that the DRG-based payment system did successfully improved the access to expensive cancer treatment. Please elaborate more on policy implications of the study results.  Any restriction or limitation for the take up of the DRG-based payment scheme?  Any influence on cancer types? 
  2. Is the financial protection/support in Chile comparable to other developed countries? Any policy recommendations?

Reviewer 2 Report

Here is my key comments.

  • The introduction section is terribly short. Please include a number of studies regarding this country and some similar countries. Further, please summarize the key similarities and difference between your study and other studies in the introduction section.
  • Please include a paragraph of your key findings in the introduction section. 
  • Please clearly explain the empirical model in the main section, not as supplementary section. Also, provide enough justification for the execution of this model. The log-linear model is a restricted form and please execute a Box-Cox transformation as the log form is nested within Classic Box-Cox Model. 
  • Regardless of any model type, you have to make sure the functional form specification is correct. In addition, how do you address the sample selection bias? Apparently, you have only included private insurance holder in the analysis. 
  • Also, please compare some of your estimates with previous studies on the same country or comparable countries. 
  • Your findings should support your conclusions. 

Round 2

Reviewer 2 Report

No further comments.